# Enhanced Antibiofilm Effects of N_2_ Plasma-Treated Buffer Combined with Antimicrobial Hexapeptides Against Plant Pathogens

**DOI:** 10.3390/polym12091992

**Published:** 2020-09-01

**Authors:** Bohyun Kim, Hyemi Seo, Jin Hyung Lee, Sunghyun Kim, Won Il Choi, Daekyung Sung, Eunpyo Moon

**Affiliations:** 1Department of Biological Science, College of Natural Sciences, Ajou University, 206, World cup-ro, Yeongtong-gu, Suwon, Gyeonggi-do 16499, Korea; amberbhk@ajou.ac.kr (B.K.); shm0827@ajou.ac.kr (H.S.); 2Center for Convergence Bioceramic Materials, Korea Institute of Ceramic Engineering and Technology, 202, Osongsaengmyeong 1-ro, Osong-eup, Heungdeok-gu, Cheongju, Chungbuk 28160, Korea; leejinh1@kicet.re.kr (J.H.L.); shkim0519@kicet.re.kr (S.K.); choi830509@kicet.re.kr (W.I.C.)

**Keywords:** antimicrobial peptides, antibiofilm, N_2_ plasma-treated buffer, pathogenic bacteria, bacterial suppression

## Abstract

Suppression of pathogenic bacterial growth to increase food and agricultural productivity is important. We previously developed novel hexapeptides (KCM12 and KCM21) with antimicrobial activities against various phytopathogenic bacteria and N_2_ plasma-treated buffer (NPB) as an alternative method for bacterial inactivation and as an antibiofilm agent of crops. Here, we developed an enhanced antibiofilm method based on antimicrobial hexapeptides with N_2_ plasma-treated buffer against plant pathogens. Our results demonstrated that hexapeptides effectively inhibited the growth of *Pseudomonas syringae* pv. *tomato* DC3000 (*Pst* DC3000) and the biofilm it formed. Potent biofilm formation-inhibiting effects of hexapeptides were observed at concentrations of above 20 µM, and samples treated with hexapeptide above 100 µM reduced the ability of the bacteria to produce biofilm by 80%. 3D confocal laser scanning microscopy imaging data revealed that the antimicrobial activity of hexapeptides was enough to affect the cells embedded inside the biofilm. Finally, combination treatment with NPB and antimicrobial hexapeptides increased the antibiofilm effect compared with the effect of single processing against multilayered plant pathogen biofilms. These findings show that the combination of hexapeptides and NPB can be potentially applied for improving crop production.

## 1. Introduction

Controlling biofilm formation by numerous plant-related pathogenic bacteria is one of the most difficult challenges faced when increasing the production of high-quality crops. Among the various methods used, antibiotics are one of the most important antimicrobial substances used in the treatment and prevention of bacterial infections. However, because of the increasing resistance to antibiotics, the types of antibiotics that can be applied is rapidly decreasing. Furthermore, in agricultural crops, cases in which microbial antibiotic susceptibility is reduced and, thus, pathogens cannot be controlled, have been reported, resulting in a decrease in yield or food poisoning in the case of crops that are eaten raw. Accordingly, the demand for the possibility of antibiotic replacement continues to increase [1].

Antimicrobial peptides (AMPs) are new antimicrobial molecules that can replace antibiotics and conventional chemicals [2,3]. AMP is a cationic and amphiphilic peptide of 10 kDa or less that acts as an important component in the natural defense against pathogen invasion in most living organisms, including plants, animals, and bacteria, and varies in length, sequence, and structure [4]. AMPs interact with negatively-charged biological membranes and have a broad spectrum of activity and low cytotoxicity against eukaryotic cells [5]. However, the practical application of AMPs has certain drawbacks including the long-term endeavors required for screening and characterization, low-level of activity, and poor bioavailability. As an alternative, positional scanning of mixture-based synthetic combinatorial libraries (SCLs) of tens of millions of peptide sequences can be performed to enable successful identification of AMPs with unprecedented or novel properties in a brief period of time [6]. The screening of a SCL built around an amphipathic 18-mer peptide sequence known to show antimicrobial activities revealed a 10-fold enhancement [7].

We previously developed four novel hexapeptides with antimicrobial activities against various phytopathogenic bacteria. KCM12 (KWRWIW-NH_2_) and KCM21 (KWWWRW-NH_2_), which were the most effective AMPs, significantly exhibited bactericidal activities against *Pseudomonas syringae* pv. *tomato* DC3000 (*Pst* DC3000). *Pst* DC3000, which forms a bacterial speck in tomatoes and other plants, is a ubiquitous environmental bacterium that can form biofilms on surfaces of crops as a survival strategy [8]. Killing bacteria in biofilms is much more difficult than killing individual cells because of the protection afforded by the extracellular polymeric substance and their multilayer structure [9,10,11]. Therefore, biofilm-associated infections result in systemic disease because biofilm-associated microbes are more resistant to antimicrobial treatment than free-living bacteria [10,12]. In addition, we previously developed N_2_ plasma-treated buffer (NPB) as an alternative method for bacterial inactivation and as an antibiofilm agent of crops. Use of NPB is advantageous over direct plasma application owing to more convenient and efficient storage and transportation. Analysis of scavenger assays with a multitude of antioxidants disclosed that reactive oxygen species contribute to the inhibitory cellular actions of NPB, with hydrogen peroxide (H_2_O_2_) and singlet oxygen (^1^O_2_) proving vital in bacterial death.

Combination treatment is being increasingly employed to enhance the antimicrobial effects against multiple potential pathogens. Therefore, in this study, we developed an enhanced antibiofilm method on the basis of antimicrobial hexapeptides with NPB against the plant pathogen *Pst* DC3000. The effect of combination treatment of hexapeptides with NPB was increased in controlling pathogenic bacteria. In addition, we analyzed the optimum combination conditions of antibiofilm efficiency and have detailed the results here. Our results show that using antimicrobial hexapeptides in combination with NPB has an excellent antibiofilm effect against multilayered plant pathogen biofilms, and that the penetration efficacy of NPB into multilayered biofilms is one of the foremost vital properties contributing to its strong antibiofilm activity.

## 2. Materials and Methods

### 2.1. Bacterial Culture and Biofilm Cultivation

Bacterial strain *Pseudomonas syringae* pv. *tomato* DC3000 (*Pst* DC3000) was inoculated in Luria-Bertani (LB) broth and incubated for 24 h at 28 °C on a shaker. The exponentially growing bacterial culture was diluted 1:100 in fresh LB medium supplemented with 0.5% glucose to form a biofilm. An aliquot containing 1 mL of diluted bacterial culture (approximately 10^9^ CFU/mL) was added to the wells of 12-well plates with 12 mm Ø microscopic cover glasses and incubated statically for 24 h at 28 °C. The liquid medium at the top of the biofilm was replaced every 6 h with care taken to ensure that bacteria were not taken along with the medium. Subsequently, the medium was removed, and the attached cells were washed three times with sterile phosphate buffered saline (PBS). The remaining attached cells were incubated for 10 min at 60 °C and stained with 0.1% (*w*/*v*) crystal violet (CV) for 15 min at 25 °C. Excessive CV stain was washed off with sterile PBS. Ethanol and acetone (95:5, *v*/*v*, 0.5 mL) were added to each well to dissolve the CV stain, and absorbance was measured at 570 nm [13,14].

### 2.2. Selection and Purification of Hexapeptides

The hexapeptides, KCM12 and KCM21, were purchased from Peptron Co. (Daejeon, Korea). The two hexapeptides selected in this study contained several hydrophobic/aromatic amino acid residues and one or two positively charged residues; their composition profiles agreed with those of many previously reported AMPs [2,3]. Each peptide was purified by preparative reversed-phase high-pressure liquid chromatography with a Shiseido Capcell Pak C18 column (Tokyo, Japan) [15]. The peptide identity was confirmed by mass analysis with a HP 1100 series LC/MSD, after which the peptides were dissolved in 50% dimethyl sulfoxide (DMSO) and stored in aliquots at −20 °C. The final concentration of the stock solution was 5 mM. 

### 2.3. Antibiofilm Test Using the Hexapeptides

Bacterial strains were cultured overnight in LB media and inoculated in fresh LB broth at a volume of 1/100. Cells were grown for approximately 6 h and harvested by centrifugation at 4000 rpm for 10 min. Cells were then re-suspended in PBS and diluted to approximately 10^7^ CFU/mL. Then, 20 µL hexapeptides, either KCM12 or KCM21, were mixed with 180 µL cell suspension. The mixed suspension was incubated for 2 h before serial dilution. Aliquots of 100 µL of diluted suspension were spread on the surface of LB agar plates. Cell suspension of 50% DMSO was equally diluted, and 100 µL aliquots were spread on the surface of LB agar plates. Plates were incubated overnight. Next, the percentage of surviving cells was calculated by colony counting. Five different concentrations (5, 10, 50, 100, and 250 µM) of KCM12 and KCM21 were applied to bacterial biofilms cultured for 24 h, and the samples were incubated for 2 h at 25 °C. After treatment, probe-labeled biofilm images were obtained by fluorescence microscopy.

### 2.4. Generation and Treatment of NPB for Enhanced Antibiofilm

NPB was generated by a micro-plasma jet device, and the generated NPB was used for antibiofilm test, as previously reported [16,17]. To generate NPB, 1 mL of PBS was added to the wells of 12-well plates, and plasma was generated using N_2_ gas with the nozzle located 1 cm above the PBS solution. A total of 250 µL of NPB was added to an equal volume of bacterial culture supernatants (10^7^ CFU/mL) for 20 min at 25 °C. All supernatants were removed from the bacteria treated with hexapeptides and treated with additional NPB (Figure 1).

### 2.5. LIVE/DEAD Bacterial Viability Assay

The biofilm was developed on 12-mm Ø microscopic cover glasses as described above, covered with 300 µL of SYTO9/PI solution (LIVE/DEAD BacLight bacterial viability assay kit, Invitrogen Co., Carlsbad, CA, USA), and incubated at room temperature for 15 min in the dark. The biofilm-containing coverslips were conveyed onto a glass slide and observed under a fluorescence microscope (Zeiss Axioscope 2, Carl Zeiss, Jena, Germany) equipped with a GFP and rhodamine filter set (X600). SYTO9/PI staining is used to determine the killing efficiency of a substance with an unknown number of dead cells. Therefore, a standard curve was created, and the ratio of live/dead cells was calculated using SYTO9 (green) to PI (red) fluorescence ratio.

### 2.6. Confocal Laser Scanning Microscopy (CLSM)

The laser was used at 488-nm excitation, and the emission was observed at 528 nm (SYTO9) and 645 nm (PI). Zeiss ZEN 2012 software was used to acquire images from nine sections of biofilms, whereas the Z-stack and CellProfiler software (Carl Zeiss, Jena, Germany) were used to analyze signal intensities and produce 3D images.

### 2.7. Statistical Analysis

The data were obtained from three independent experiments and analyzed using analysis of variance test. Treatment groups were analyzed by ANOVA using SPSS 22.0, and a probability (*p*) value of <0.05 was considered statistically significant. All the experiments were repeated at least three times, and statistical significance was determined using one-way ANOVA (Dunnett T3) (* *p* ≤ 0.05, ** *p* ≤ 0.01, and *** *p* ≤ 0.001).

## 3. Results and Discussion

### 3.1. Inhibitory Efficacy of Hexapeptide Against Plant Pathogenic Bacteria

The optimal treatment conditions were established by comparing the antibacterial effects against pathogens and hexapeptide concentrations (Figure 2). The colony counting assay determined the number of *Pst* DC3000 surviving populations. KCM12 (Figure 2A) and KCM21 (Figure 2B) treatment effectively reduced bacterial populations in the *Pst* DC3000 suspension. Five different concentrations (10, 20, 100, 250, and 500 µM) of hexapeptides KCM12 and KCM21 were applied against planktonic *Pst* DC3000. As shown in Figure 2, increased concentration of hexapeptide showed a strong antibacterial effect on pathogens. In the presence of high concentrations of hexapeptides (above 250 µM), a strong inhibition activity was observed. The hexapeptides demonstrated high antimicrobial activity against pathogenic bacteria and were considered to be a promising alternative to previous antibiotics [18,19]. We proved that the novel hexapeptides, KCM12 and KCM21, exhibit broad-spectrum antimicrobial activity.

### 3.2. Antibiofilm Effect of Hexapeptides Against Pst DC3000 Biofilm

*Pst* DC3000 is a plant pathogen that causes bacterial specks in tomatoes and other plants. In chronic infection, the bacteria demonstrate adaptive behaviors that increase their resistance to antibiotics, including the expression of genes associated with biofilm formation. Biofilms of *Pst* DC3000 have a multilayer structure; therefore, it is more challenging to kill microorganisms in a biofilm than individual cells. Figure 3 shows some hexapeptides and their antibacterial effects against biofilms of *Pst* DC3000. Four different concentrations (10, 20, 100, and 250 µM) of hexapeptides were applied to *Pst* DC3000 biofilms. CV staining showed inactivation of *Pst* DC3000 biofilm upon treatment with the various hexapeptide concentrations. The antibiofilm effect of hexapeptides on *Pst* DC3000 was dose-dependent, and a strong antibiofilm effect was observed at concentrations of more than 100 µM. The inhibitory effects of the two hexapeptides were similar. KCM12 and KCM21 are very similar in terms of amino acid composition, and both hexapeptides showed high activity against *Pst* DC3000. CV staining was used to monitor biofilm formation capacity, and the biofilm mass was measured at 570 nm on a microplate reader (Figure 3C). Each absorbance value was calculated by subtracting the means of absorbance of a blank sample (untreated PBS) [14]. The absorbance values of both KCM12 and KCM21 demonstrated that 20 µM of hexapeptide treatment reduced the ability of the bacteria to produce biofilm by 20%. Compared to those of samples without hexapeptide treatment, the optical density (OD) values of both KCM12- and KCM21-treated samples decreased in a concentration-dependent; in particular, the OD values decreased by a large margin for the 100 µM and 250 µM concentrations compared with those for the 10 µM and 20 µM concentrations. In addition, there was no significant difference in the anti-biofilm effect between the 100 µM and 250 µM concentrations of hexapeptides. In other words, potent biofilm formation-inhibiting effects of hexapeptides were observed at concentrations of 100 µM or more. Samples treated with 20 µM or more of hexapeptides had a significantly lower absorption strength than the untreated control, and statistical analysis also confirmed significant differences.

### 3.3. Verification of Antibacterial Effects Against Pst DC3000 by LIVE/DEAD Assay

We analyzed the images of the AMP-mediated antibacterial effects against *Pst* DC3000 using fluorescence microscopy (Figure 4) and results of the LIVE/DEAD assay. To determine the inhibitory effect of the hexapeptides, samples were treated with various concentrations of KCM12 and KCM21. Both hexapeptides KCM12 and KCM21 showed a strong inhibitory effect at concentrations of 100 µM or more. Our results showed that following treatment with 250 µM hexapeptides, the number of dead cells increased by more than 80% compared to that of the untreated sample. We assessed the inhibitory activity of hexapeptides against biofilm generation by *Pst* DC3000 using images from the fluorescence microscopy, which clearly showed the strong inhibition abilities of KCM12 and KCM21 on the biofilm formation by *Pst* DC3000. A comparison of the inhibitory effects of the two hexapeptides showed that KCM21 had a stronger biofilm inhibition effect than did KCM12. Regardless of this slight difference in biofilm formation inhibition between the hexapeptides, both hexapeptides were very effective in controlling biofilm formation. Considering the difficulty reported in the inhibition of biofilm-forming bacteria because of the protection imparted by their biofilms [20,21], the ability of hexapeptides to inhibit *Pst* DC3000 is promising. Thus, hexapeptides have the potential for use as novel antibiofilm or antimicrobial agents for prevention or treatment of biofilm-related infections.

### 3.4. D Analysis on Antibiofilm Efficacy of Hexapeptides Using CLSM

The inhibition efficacy on the biofilms formed by *Pst* DC3000 was further examined using serially diluted hexapeptides. To obtain the final 3D CLSM images in Figure 5, each image from nine optical sections, positioned at different depths in the biofilm, was acquired as 3D datasets. Z-stacks of nine images were rendered into the 3D mode and visualized from three different angles (three rows in Figure 5A,B). To determine the inhibitory effect of the hexapeptides, samples were treated with various concentrations (20, 100, and 250 µM) of KCM12 and KCM21 (X630). Compared with the inhibition efficiency against *Pst* DC3000, which was 100% at up to 250-µM concentration of hexapeptides, as shown in Figure 2, the inhibition activity against the resistant biofilm was slightly less, but was well maintained up to a concentration of 250 µM. The total red intensity of the 3D images was determined using the Zeiss ZEN 2012 software, and the results are shown in Figure 5C. The antibiofilm activity of hexapeptides was clearly confirmed using red images and the total red intensity, which indicate the dead bacterial cells. CLSM images of biofilms treated with hexapeptides at a 250-µM concentration displayed high intensity of red fluorescence; in contrast, the untreated control image exhibited completely green fluorescence. Compared with the untreated control, there was a significant difference in the red fluorescent intensity values. The result confirmed that the value of red fluorescence intensity was increased by more than 10% at 100 µM hexapeptide concentrations and by more than 30% at 250 µM hexapeptide concentrations. Bacteria embedded inside biofilms can survive the inhibitory action of most antibiotic agents, as the cells are secured by the barrier made by the biofilm [16]; however, hexapeptides effectively inhibit cell growth of cells exposed on the surface of the biofilm [9,22]. The 3D analysis revealed that hexapeptides effectively inhibited bacterial cells not only near the surface of the biofilm but also deeply inside the biofilm, suggesting that the inhibitory agent had high penetration abilities [17]. In this study, KCM12 and KCM21 showed high activity against *Pst* DC3000 biofilm formation in a dose-dependent manner. There are several different classes of antibacterial peptides, including those rich in a particular amino acid [23]. Among them, peptides rich in Arg and Trp residues are very important and have broad and potent antibacterial activities [23]. Positional scanning of synthetic combinatorial libraries (PS-SCLs) of hexapeptides revealed that these two amino acids are involved in the antibacterial activity. In previous studies, these peptides showed no cytotoxic effects in the host and were associated with a low risk of occurrence of a resistant strain. These novel hexapeptides can be considered to be alternatives to conventional antibiotics.

### 3.5. Enhanced Antibiofilm Effects of NPB Combined with Hexapeptides

Several novel antibiotic combinations demonstrate increased activity compared with that of single agent. NPB, developed as an alternative method for bacterial inactivation and biofilm suppression, was combined with hexapeptides to maximize the antibacterial effect. *Pst* DC3000 cultured in liquid medium was treated with a combination of NPB and hexapeptide (Figure 6A). After treatment with NPB (five-fold dilution and 10-fold dilution) and hexapeptide (20 and 100 µM), the results showed a significant antimicrobial effect. When compared with the result of only hexapeptide treatment, combination treatment exhibited enhanced inhibitory effects. Results of combination treatment for *Pst* DC3000 biofilm inhibition were confirmed by CLSM (Figure 6B). Three-dimensional images showed that the inhibitory effects of combination treatment were much more effective than those of the hexapeptide only. After treatment with 100 µM hexapeptide and NPB (five-fold dilution), an enhanced antibiofilm effect was observed. Biofilm treated with combination treatment had most of the cells stained red, indicating that the cells were dead. Compared with controls treated with NPB or hexapeptide alone, there was a significant difference in the red fluorescent intensity. Based on these results, combination treatment of NPB and antimicrobial hexapeptides improved the antibiofilm effect when compared with the effect of treatment with a single agent against multilayered plant pathogen biofilms. In addition, with the combination of NPB and hexapeptide, KCM21 was more effective than KCM12. KCM12 and KCM21 are very similar in terms of amino acid composition and showed high activity against *Pst* DC3000. However, KCM21 has a higher hydrophobicity as well as strong inhibition activity [15]. We considered that the difference in hydrophobicity between the peptides may have had a slight effect on biofilm control in combination treatment.

## 4. Conclusions

Suppression of pathogenic bacterial growth is an important part of molecular biological and biochemical research and for microbial control in the food and agricultural industries. In this study, two hexapeptides (KCM12 and KCM21) whose antibacterial effects were verified on plant pathogens were selected to confirm their antimicrobial activity against *Pst* DC3000. Furthermore, NPB, developed as an alternative for bacterial inactivation and biofilm suppression, was combined with the hexapeptides to enhance the antimicrobial effect. CLSM images showed that the antibiofilm effects of combination treatment were much more effective than those of the hexapeptide or NPB alone. Three-dimensional analysis revealed that combination treatment effectively inhibited bacterial cells embedded deep within the biofilm as well as near the surface of the biofilm, suggesting that the inhibitors were highly penetrative. These findings suggest that the combination of hexapeptides and NPB can be potentially applied to increase yield and expand the shelf life of food for improving crop production.

## Figures and Tables

**Figure 1 polymers-12-01992-f001:**
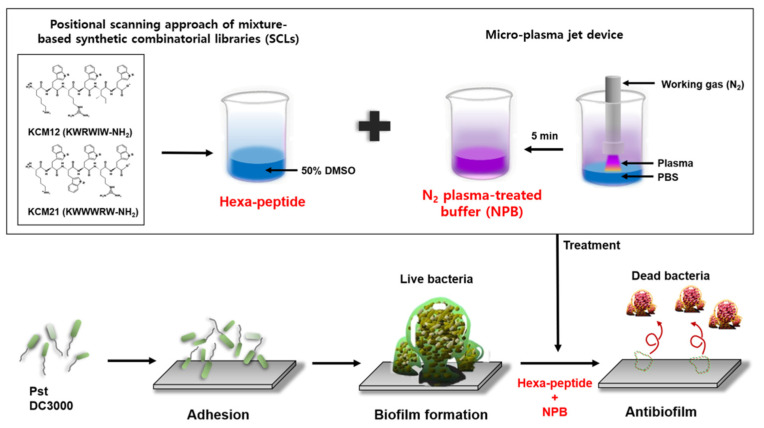
A schematic diagram depicting the experimental design of hexapeptide (KCM12 and KCM21) and N_2_ plasma-treated buffer (NPB), generation of *Pst* DC3000 as plant pathogenic bacterium, and enhanced antibiofilm effects.

**Figure 2 polymers-12-01992-f002:**
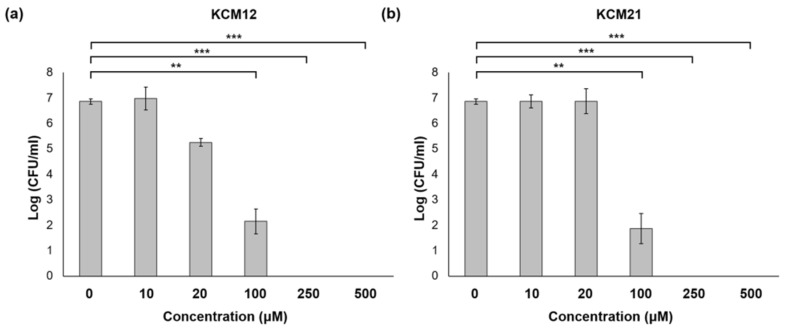
Antibacterial efficacy of hexapeptides (**a**) KCM12 and (**b**) KCM21 against *Pst* DC3000 under various concentration, (** significant at *p* ≤ 0.01 between control (0 μM) and KCM12- or KCM21-treatment and *** significant at *p* ≤ 0.001).

**Figure 3 polymers-12-01992-f003:**
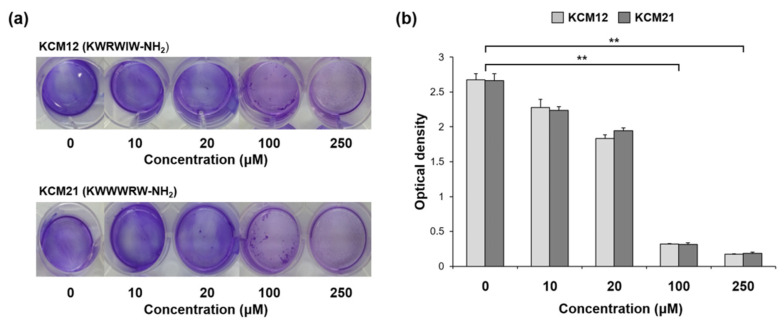
**Antibiofilm effect of hexapeptides against *Pst* DC3000 biofilm** (**a**) Images of biofilm formation capacity using crystal violet (CV) staining and (**b**) biofilm mass measurements after hexapeptide treatment of *Pst* DC3000, (** significant at *p* ≤ 0.01 between control (0 μM) and KCM12- or KCM21-treatment).

**Figure 4 polymers-12-01992-f004:**
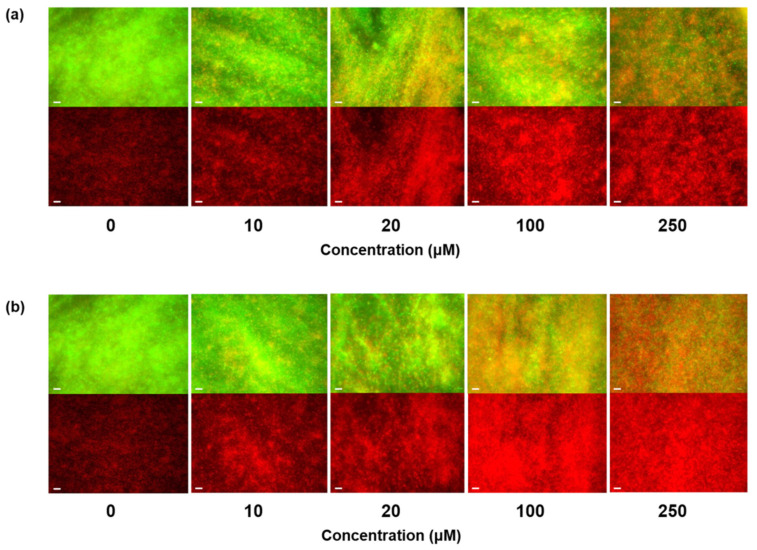
Fluorescence microscopy images using LIVE/DEAD bacterial viability assay, showing the inhibitory effect of (**a**) KCM12 and (**b**) KCM21 on biofilms of *Pst* DC3000. Scale bar represents 20 μm.

**Figure 5 polymers-12-01992-f005:**
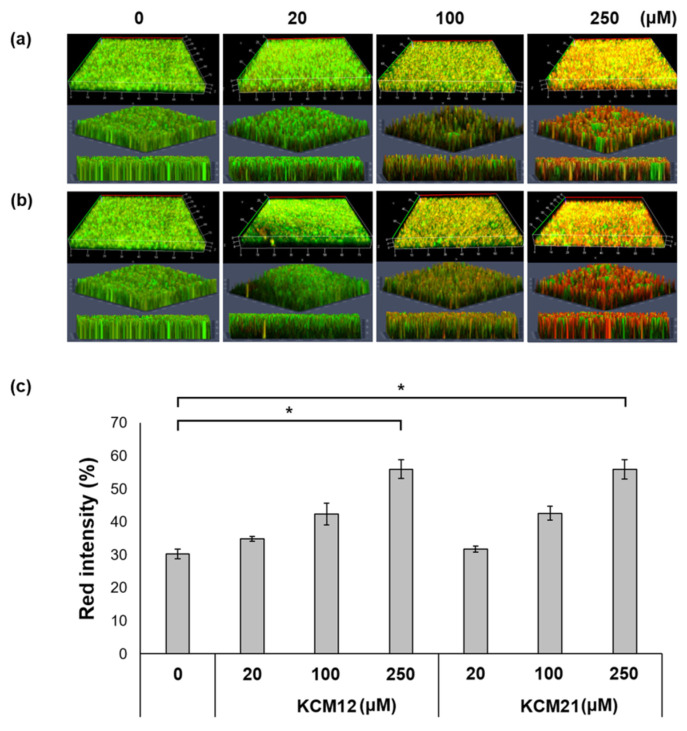
CLSM imaging analysis showing biofilm formation of *Pst* DC3000 after (**a**) KCM12 and (**b**) KCM21 treatment. 3D CLSM images (X630) treatment as Z-stacks of the nine images at three different angles, and (**c**) red intensity profile as the set of intensities after hexapeptide treatment. (* significant at *p* ≤ 0.05 between control (0 μM) and KCM12- or KCM21-treatment).

**Figure 6 polymers-12-01992-f006:**
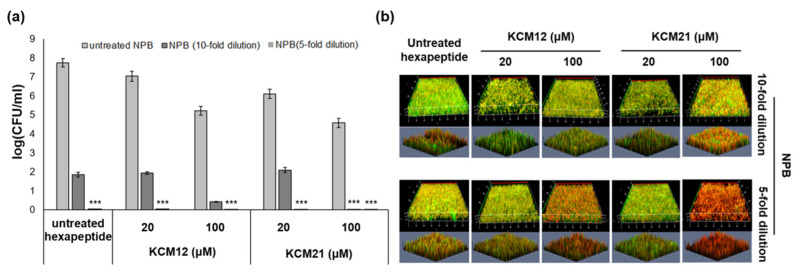
Evaluation of antibacterial and antibiofilm effect of combined treatment with NPB and hexapeptide in (**a**) free-living bacteria, and (**b**) biofilms of *Pst* DC3000. (*** significant at *p* ≤ 0.001 between control (0 μM) and KCM12- or KCM21-treatment).

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
