# Peer review of "Enhanced Antibiofilm Effects of N2 Plasma-Treated Buffer Combined with Antimicrobial Hexapeptides Against Plant Pathogens"

_polymers, 2020, doi:10.3390/polym12091992_

Round 1

Reviewer 2 Report

The combination treatment of P. syringae formed biofilms using N2 plasma-treated buffer (NPB) and antimicrobial hexapeptides is analyzed in this manuscript. Increased bacterial inactivation was observed when compared to the effects of single agents.  It was concluded that “effect of combination treatment using hexapeptides and NPB is exceptionally synergistic in controlling pathogenic bacteria”. To my mind, there are no evidences provided to call this interaction as synergy and only P. syringae cells were analyzed

The main problem is a lack of novelty in this study. There is a paper published on antibacterial action of hexapeptide KCM21 on P. syringae cells ([2] in References), there is another paper on “Super-antibiofilm effect of NPB against plant pathogenic bacterium (mostly about P. syringae, ref [16]). What is new in this manuscript? According to Conclusions – “The combination treatment of NPB and antimicrobial hexapeptides significantly increased bacterial inactivation when compared to the effect of a single agent”. This is interesting, but only one figure and 11 lines of text! I think that this is not enough. “Three-dimensional analysis revealed that combination treatment effectively inhibited bacterial cells embedded within the biofilm, suggesting that the inhibitors were highly penetrative”, but similar conclusion we can find in the description of NPB alone: “The result clearly showed effective penetration of the inhibitory activity through biofilm layers, and thus provided the cellular basis for the super-antibiofilm activity of NPB” [16].

Some additional remarks:

  1. The optimum temperatures for growth of syringae is 20–28 °C, 37 °C is very high! After 24 h of growth in LB medium the culture should not be exponential, but stationary!
  2. What is the rationale to add glucose into LB medium? Glucose cannot get into syringae cytoplasm (there are no transporters for glucose in P. syringae plasma membrane) and cannot be used for synthesis of exopolysaccharides.
  3. Amount (initial concentration?) of bacteria transferred to plates for biofilm formation is not clear
  4. Not clear, what concentrations of hexapeptides were used in experiments presented in Figure 2: as shown in the figure or as stated in the text.
  5. Figure 2 does not present any new results: 1) hexapeptide KCM21 was already described in [2] and 2) through all this study no considerable difference in activity between KCM12 and KCM21was detected. What was the difference in structure of these peptides? Why these peptides were selected?
  6. 3. This figure has no title?
  7. Not good to write in two ways: “hexapeptide” and “hexa-peptide”.
  8. In Fig. 6(a) we find KCM12 twice, one should be KCM21
  9. Too long introductory part in CONCLUSIONS, not enough information about the results
  10. Meaning explanation of * and ** in the Figures is not presented

And one question: why this manuscript was submitted to Polymers?

Round 2

Reviewer 1 Report

The statistical analysis should be done through analysis of variance (ANOVA) using SPSS 13.0 software. Duncan's multiple range tests were used to determine significant difference among mean values. For example, In figure 3b, the differences between '10 um and 100 um' or '10 um and 250 um' or '20 um and 100 um' or '20 um and 250 um' were not found in the 'results and disscusion' section.

Reviewer 2 Report

  1. Most of abstract readers will not understand “Pst DC3000”. It should be clearly written that this is syringae pv. tomato DC3000.
  2. Now is clear that bacterial cultures were grown at 28 oC, not at 37 o I agree, that effect of glucose on P. syringae cells can be indirect, through the expression of external polisaccharade-related genes. However, the protocol for biofilm cultivation still needs a special attention. 24 h of cultivation without aeration in the closed wells, in the glucose-containing medium, starting concentration 1x109 cells/ml. These are anaerobic (or microaerobic) conditions, but P. syringae are aerobic bacteria, they need oxygen. Such conditions are good for fermenting bacteria and 24 h is long enough period to get mixed biofilms. How the composition of the obtained biofilms was controlled?
  3. p. 8. “KCM21 has a higher hydrophobicity as well as higher activity”. – Some reference or data about the hydrophobicity of the hexapeptides needed to justify this
  4. p. 9: “In this study, two hexapeptides (KCM12 and KCM21) with excellent antibacterial effects on plant pathogens were selected, and their antimicrobial activity was significantly enhanced against Pst DC3000.” – I do not think, that “antibacterial effects were excellent”, if it was possible to “significantly enhance” them.
  5. p. 9. “After treatment with NPB and hexapeptide, the results showed a significant synergistic antibiofilm effect”. – I still do not see any evidence of synergy between hexapeptides and NPB. For this FIC (comparing the MICs of each agent alone with the combination-derived MIC) or similar characteristics should be provided.
  6. What is the price of the peptides obtained from Peptron Co.? How much would it cost to treat 1 hectare of crops with 0.1 mM solutions of hexapeptides?

Round 3

Reviewer 1 Report

Ok

Reviewer 2 Report

Authors properly replied to all my remarks, the manuscript has been improved.  I still have some doubts about the novelty of the paper, but quality of the manuscript is good enough to be published.